# Implications of Birth-Dose Vaccination against Hepatitis B Virus in Southeast Asia

**DOI:** 10.3390/vaccines9040374

**Published:** 2021-04-12

**Authors:** Sheikh Mohammad Fazle Akbar, Mamun Al Mahtab, Ferdousi Begum, Shaikh A. Shahed Hossain, Sukumar Sarker, Ananta Shrestha, Md. Sakirul Islam Khan, Osamu Yoshida, Yoichi Hiasa

**Affiliations:** 1Department of Gastroenterology and Metabology, Ehime University Graduate School of Medicine, Toon City, Ehime 791-0295, Japan; yoshidao@m.ehime-u.ac.jp (O.Y.); hiasa@m.ehime-u.ac.jp (Y.H.); 2Department of Hepatology, Bangabandhu Sheikh Mujib Medical University, Dhaka 1000, Bangladesh; shwapnil@agni.com; 3Holy Family Red Crescent Medical College, Dhaka 1000, Bangladesh; smbakhtiar2017@gmail.com; 4BRAC James P. Grant School of Public Health BRAC University, Dhaka 1212, Bangladesh; shahed.hossain@bracu.ac.bd; 5Save the Children Fund, Dhaka 1212, Bangladesh; ssarker56@gmail.com; 6Liver Foundation Nepal, Kathmandu 44600, Nepal; anant_02@hotmail.com; 7Department of Anatomy and Embryology, Ehime University Graduate School of Medicine, Toon City, Ehime 791-0295, Japan; sakirul@m.ehime-u.ac.jp

**Keywords:** hepatitis B vaccine, EPI, birth-dose vaccine, SEARO

## Abstract

The World Health Organization (WHO) South-East Asia Regional Office (SEARO) covers 11 countries with a combined population of about 2 billion people, making it the most populous of the six WHO regions. In 1992, the WHO advocated including the hepatitis B vaccine in the Expanded Program of Immunization (EPI) and vaccinating all infants and children three times within 1 year of birth (HepB3). Recently, the WHO advocate birth-dose hepatitis B vaccination (HepB-BD) as soon as possible after birth, preferably within 24 hours. In 2016, the SEARO endorsed a regional hepatitis B control goal with a target of hepatitis B surface antigen (HBsAg) seroprevalence of ≤1% among children aged ≥5 years by 2020. Of the 11 SEARO countries, four achieved this target on schedule. Out of these four countries, two countries (Bangladesh and Nepal) have not adopted HepB-BD in EPI program. On the other hand, the coverage of HepB3 is not satisfactory in some SEARO countries, including India which adopted HepB-BD but could not achieve the overall target of SEARO. Thus, it is a point of debate whether emphasis should be placed on proper implementation of HepB3 or whether a new agenda of HepB-BD should be incorporated in developing countries of SEARO. The article discusses strengthening and expanding the Hepatitis B vaccination program in SEARO countries with an emphasis on HepB and HepB-BD programs.

## 1. Introduction

Hepatitis B virus (HBV) infection is a major global public health problem and about 2 billion people are infected by HBV at some point in their lifetime [1]. Of these, about 257 million are chronically infected with HBV, a condition that is characterized by the expression of the hepatitis B surface antigen (HBsAg) and HBV DNA in the serum, with or without elevated alanine aminotransferase. Chronic HBV carriers are primarily responsible for maintaining the HBV transmission cycle. A considerable proportion of people with chronic HBV infection eventually develop complications like liver cirrhosis and hepatocellular carcinoma, which is responsible for about 887,000 deaths per year worldwide [2]. Therefore, HBV infection represents a major global public health problem, although HBV-related hazards and complications are most evident in developing countries in sub-Saharan Africa, Asia, and Latin America.

The currently available antiviral drugs cannot completely eradicate HBV in people with chronic HBV infection. These drugs are endowed with several limitations, such as cost, infinite usage or usage for life, development of mutant strains, and inability to contain progression of liver diseases [3,4,5]. Accordingly, preventing HBV infection has become the principal target for containing the infection. Several preventive measures have been adopted at national and regional levels in different countries. The prevention of HBV infections is dependent on various factors (1) the administration of three doses of hepatitis B vaccine (HepB) for infants, (2) better management of HBV blocking from mother to child by providing birth dose vaccine (3) routine testing and treatment of pregnant women, (4) blood, injection and surgical safety, and (5) harm reduction for people who inject drugs. Although there are several complicated means to prevent HBV transmission, prevention is documented as being cost–effective [6,7,8,9].

The World Health Organization (WHO) initiated the Expanded Program on Immunization (EPI) in May 1974 with the goal of vaccinating children worldwide [7,9]. The target was to provide universal immunization for all children by 1990, as this was considered essential for the WHO strategy to achieve “health for all” by 2000. The diseases targeted by the EPI included diphtheria, whooping cough, tetanus, measles, poliomyelitis, and tuberculosis. In 1992, the WHO recommended that HB vaccination be included in the EPI program, although Thailand implemented it in 1988 [8]. By the end of 2014, 168 countries had adopted universal HB immunization within the framework of the EPI program. In the program, all infants and children are vaccinated three times with HB vaccine (HepB) to ensure prolonged protection against the disease. Evidence suggests that birth-dose hepatitis B vaccination (HepB-BD) combats HBV infection more efficiently. It has been shown that the addition of HepB-BD among displaced Somali refugees in Djibouti camps would save 9807 life-years/year, with an incremental cost-effectiveness ratio of 0.15 USD (US dollars) per life-year saved when compared with traditional vaccination programs [10]. Similar cost-effectiveness and efficacy of HepB-BD was found in other African refugee camps [10]. In Ethiopia, the prevalence of HBV among pregnant women is 5%. However, adding a birth dose of HBV vaccine would present an incremental cost-effectiveness ratio (ICER) of USD 110 per disability-adjusted life years (DALY) averted. The estimated ICER compares very favorably with a willingness-to-pay level of 0.31 times gross domestic product per capita (about USD 240 in 2018) in Ethiopia [11]. The authors claimed that ICER estimates were robust over a wide range of epidemiologic, vaccine effectiveness, vaccine coverage and cost parameter inputs. Finally, the study concluded that based on their cost-effectiveness findings, introducing a birth dose of HBV vaccine in Ethiopia would likely be highly cost-effective, an important factor for resource-constrained countries. However, the implication of the study could help guide policymakers in considering including HBV vaccine birth-dose in other countries with similar socio-economic backgrounds, including most of the South-East Asia Regional Office (SEARO) countries.

The HepB-BD delivers a dose of monovalent HepB within 24 hours after birth. Many countries, especially in the Western Pacific, Americas, and Caribbean, have already incorporated the birth dose as part of the routine EPI [12]. The South-East Asia Regional Office (SEARO) covers 11 countries in a region of Southeast Asia with a combined population exceeding 2 billion, including many people with chronic HBV infection. HB vaccination started in SEARO countries in the late 1900s, but there is considerable heterogeneity in the timing of vaccination initiation, dosing, and coverage. It is important to assess the effects of HepB in the EPI in SEARO countries. Out of the 11 countries of SEARO, eight countries have adopted HepB-BD vaccination. However, proper assessments were not accomplished in some of these countries regarding the overall target of containment of HBV in these countries. Implementation of HepB-BD requires some specific facilities [13,14,15]. These include hospital delivery, abundant supply of vaccine at the periphery, and proper staff to administer the vaccines. Additionally, the mother should be convinced that it is safe to inject infants with the vaccine on the day of birth. In addition, natural calamity, local problems, and political factors remain an obstacle for implementing these new systems of disease prevention. Taking these factors into consideration, we attempted to get more insights about traditional Hep-B and HepB-BD in SEARO countries.

## 2. Attainment of the Hepatitis B Vaccination Target by SEARO Countries

In the pre-vaccination era from the 1980s to the early 2000s, the estimated global prevalence of HBV in children younger than 5 years was around 5% [16]. With vaccination and other measures, HBV infection had been reduced globally to less than 1% among children aged less than 5 years of age by 2019 [16,17]. However, it is necessary to assess the efficacy of HBV containment efforts at the regional scale, as there is marked heterogeneity among regions.

We found that the WHO goal of HBV containment has not been achieved at the global level, or even at the level of individual WHO regions, especially in children aged less than 5 years. SEARO covers 11 countries (Bangladesh, Bhutan, Democratic People’s Republic of Korea, India, Indonesia, Maldives, Myanmar, Nepal, Sri Lanka, Thailand, and Timor-Leste) with a combined population of about 2 billion living under diverse social and economic conditions. There are an estimated 40 million people in SEARO countries with chronic HBV infection and about 285,000 deaths annually from complications of the disease (e.g., liver cirrhosis and hepatocellular carcinoma) [18]. Similar to other parts of the world, in SEARO countries chronic HBV infections are usually acquired in infancy via perinatal or early childhood transmission.

In line with the WHO Global Health Sector Strategy on Viral Hepatitis 2016, the Immunization Technical Advisory Group of SEARO set a target of ≤ 1% seroprevalence of HBsAg among ≥5-year-old children by 2020. In 2019, eight regional and international experts assessed the status of HBV infection among children born in SEARO countries after implementation of nationwide HB immunization programs (either HepB3 or HepB-BD) via a nationally representative survey. The expert committee also analyzed Hep3/HepB-BD coverage at the national and regional scales. As shown in Table 1, the committee found that Bangladesh, Bhutan, Nepal, and Thailand have achieved an HBsAg prevalence of less than 1% among target group [19].

Table 2 provides more detailed information on the four countries reaching the HB vaccination goal [19].

## 3. Possible Reasons Seven SEARO Countries Missed the Hepatitis B Vaccination Target

While Bangladesh, Bhutan, Nepal, and Thailand achieved the HBsAg-positivity target for children of the target group, the other seven SEARO countries did not. It is difficult to determine the likely reasons for this, as there is a diverse range of potentially important factors, and heterogeneity among and within countries. More insight could be obtained by analyzing HBV prevalence among children. However, this requires nationwide data on HBsAg seroprevalence, which are currently lacking. India, Maldives, North Korea, Sri Lanka, and Timor-Leste have not conducted any empirical surveys, while in Indonesia and Myanmar there were problems with the surveys [18]. In India, the prevalence of HBsAg among pregnant mothers is 1–9%, and is highly variable among tribal and non-tribal peoples [20,21]. A high baseline prevalence of HBsAg positivity may be the most important barrier to achieving the HBsAg prevalence target of less than 1% among children aged below 5 years. A nationwide survey is required to determine whether any country has reached the WHO target, and most SEARO countries have not conducted any such survey within the past decade [18]. There are no data from countries that did not achieve the target of < 1.0 % of HBV in children aged under 5 years. While the percentage of vaccination coverage against HBV is important for attaining the target, the extent of coverage at the district level is equally important. Of the districts in India, Myanmar, and Timor-Leste there is >80% coverage with HepB in only 77%, 77%, and 54%, respectively. Thus, while there is a national vaccination program, many remote districts are not covered by vaccines to contain new HBV infections [19].

## 4. Implications of HepB-BD for SEARO Countries

Analyses of various factors provided more insight into the status of the HB vaccination program in SEARO countries. The year in which HepB for infants was introduced varied considerably among these countries, from 1997 to 2007. Thus, at least 13 years have elapsed since the introduction of HepB in these countries. Vaccine coverage has increased progressively in SEARO countries and currently exceeds 80% in all of them; the average is 91%, which far exceeds the global coverage rate of 45% [16,17,18]. Although Bangladesh and Nepal reached the HBV vaccination target among children, neither country has adopted HepB-BD. By contrast, North Korea, Indonesia, and Maldives adopted HepB-BD vaccination programs between 2000 and 2002, but have not attained the target of less than 1% HBsAg prevalence in children.

## 5. Re-Evaluation of HepB-BD in SEARO Countries

Only four SEARO countries adopted HepB-BD in 2016, which increased to seven by 2020. Although inclusion of HepB-BD in the EPI program is a positive step toward controlling HBV infection, the success of HepB-BD depends on the way in which it is implemented. India and Indonesia are the two most populous SEARO countries, with respective populations of 1.4 billion and 200 million people. The HepB coverage in 80% of the districts in these two countries was only 77%. A nationally representative Indian study found that less than 40% of children born after the introduction of HepB were positive for antibodies to HBsAg (anti-HBs) due to vaccination [20]. Therefore, the necessity of introducing a new system of HepB-BD, or expanding and optimizing the old HepB3 system, so that all newborns receive HepB is an important question. SEARO countries have certain socioeconomic and cultural limitations. For example, about 50% of newborn deliveries occur at home in most SEARO countries, rather than in a hospital, and HepB-BD may not be realistic for home-delivered infants. Additionally, the administration of HepB-BD within 24 hours of birth may not be practical in all SEARO countries. In many countries, HepB is given as a part of a pentavalent vaccine against diphtheria, whooping cough, tetanus and Haemophilus influenza B [22]. This is widely supported by policymakers, politicians, public health workers, doctors, and pregnant mothers, and a system is in place enabling most infants to receive the pentavalent vaccine. However, several types of preparations are required for the implementation of birth-dose vaccination [13,14,15]. In SEARO countries, the attitude of the mothers and policy makers to birth-diose hepatitis B vaccination program should be properly assessed and realized. Additionally, there should be a clear explanation regarding future doses of hepatitis B vaccination, which should be taken from different institutions. If these cannot be ascertained, there will be a negative impact on the HBV vaccination program conducted by the EPI.

## 6. Overall Limitation of Vaccination Program in SEARO Countries

The lack of proper and reliable data is one of the main obstacles for implementing any type of vaccination program in SEARO countries. Most of the countries have not accomplished timely nationwide survey. This links directly to limited access to testing. In addition, most of the countries in the SEARO region are resource-constrained. More than half the population in the region lives in countries with no provision for free testing. Additionally, some countries have not been able to implement the mandatory HBsAg screening of blood and blood products, and blood transfusion without screening is still practiced. In addition to these situations, currently, there is no government funding for treatment of chronic hepatitis B patients. In addition to access to testing, improving diagnosis requires awareness of the risks and routes of transmission among those who may have been exposed to the hepatitis B virus. In fact, this represents a part of global problem. Most of the patients with chronic hepatitis B are unaware of their HBV status. This is also crucial for prevention. However, government-funded public awareness work is not regular and more effort and resources are required. The major programs should be accomplished at the policy-making levels, but most of the government machinery is reluctant to take care of these facts. In addition, some scientific and technical problems prevail about HBV control in Asia. The concept of occult HBV infection and the utility of nucleic acid testing is yet to surface in most of these countries [23].

## 7. HBV Vaccine Escape Mutation

The target of hepatitis B virus (HBV) immunization is to induce the immune response in the host, resulting in the prevention of replication of HBV and containing of further spread of HBV. However, HBV escape mutation can alter the scenario of vaccine-induced protection from HBV infection. The concept of vaccine escape was first introduced in 1988 by Zanetti et al. [24]. In a follow-up study of childhood vaccination, it was revealed that vaccinated children with a strong antibody response to HBsAg could still become HBsAg positive. HBV escape mutation is usually caused by amino acid changes within the “a” determinant arising from selection or natural variation and can lead to conformational changes, which can affect the binding of neutralizing antibodies with several possible consequences [25]. In this context, it is important to assess the implications of HBV vaccine escape mutants in SEARO countries. One study that checked the variants within the "a" determinant of HBs gene in children and adolescents with and without a hepatitis B vaccination as part of Thailand’s EPI program sequenced 49 HBV genomes that encode the major antigenic epitopes of HBsAg. Eleven of the 49 children displayed variable mutations of the "a" determinant revealed in the region. Although they were vaccinated, these children became HBV carriers [26]. One study from India, however, found different mutations in the HBV genome, however G145 vaccine escape mutation was rare [27]. Studies from Bangladesh have shown a considerable number of the population (51%) had HBsAg escape mutation [28]. Accordingly, more studies will be required to have proper scientific insights regarding this. In fact, most of the SEARO countries lack facilities to check HBV genome analysis and other studies to develop insights about escape mutants.

## 8. Containment of HBV Transmission to Newborns: A Second Look at Pregnant Women

Vaccination against HBV with HepB or HepB-BD remains the main strategy to prevent HBV infection in neonates and children. Outcomes could be improved by paying additional attention to pregnant women during the antenatal period. Viral hepatitis has been included in the 2030 United Nations Sustainable Development Goals (SDGs), among which SDG 3.3 aims to ”combat viral hepatitis” and “eliminate hepatitis by 2030”. To attain this goal, the WHO set a target of less than 1.0% HBsAg positivity in children aged less than 5 years by 2020. To eliminate HBV by 2030, however, the HBsAg prevalence in children less than 5 years of age should be 0.1%. To achieve this goal, nine global targets must be met, including “reducing new cases of chronic viral hepatitis B infections by 90% and deaths by 65% by 2030”. The goal may be achievable by following the WHO vaccination schedule (universal immunization of infants with three or four doses of HepB, with the first dose given within 24 h after birth). At the same time, HB vaccine has recently been found to have therapeutic effects and these are now under extensive clinical trials. New and novel candidates for therapeutic vaccines have been explored and phase I/II/ and III clinical trials with those vaccines have been accomplished [29,30,31]. Antenatal care should be optimized by checking all pregnant mothers for HBsAg, assessing HBV DNA and hepatitis B e antigen (HBeAg), quantifying HBV DNA, and giving antivirals to HBsAg-positive mothers with high HBV DNA levels [32,33]. It is true that these antenatal care approaches cannot be optimized in all countries of SEARO and not even in all parts of a country. The facilities for the assessment of HBV infection and HBeAg positivity are not common in most SEARO countries. In addition, the quantification of HBV DNA cannot be accomplished in most of the SEARO countries. There is no national health insurance policy in most of these countries for majority of the people. Thus, the proper evaluation of mothers about HBV infection is a tough job [34,35,36,37]. However, antenatal care represents an effective mode of preventing mother to child transmission of HBV from pregnant women to their newborns within the perinatal period. At the same time, the coverage of Hep-B has reached more than 80% in 10 SEARO countries and eight countries have adapted HepB-BD. After accomplishing these, it is the time to introduce the concept of antenatal care for prevention of HBV in infants and child. This will also positively influence the concept of “Elimination of Hepatitis by 2030”.

## 9. Conclusions

HBV is endemic in SEARO, with neonatal and perinatal HBV transmission being a major public health concern. To contain HBV, neonates are immunized with HepB in all 11 SEARO countries. The target of this vaccination program is to reduce HBsAg-positivity to < 1.0 % among children aged less than 5 years. This target has been achieved by four of the 11 SEARO countries: Bangladesh, Nepal, Bhutan, and Thailand. Bangladesh and Thailand have respective populations of 164 and 70 million people, while Nepal and Bhutan are smaller (29 and 0.7 million, respectively). The success of HB vaccination in Bangladesh, a comparatively small country with a huge population and a moderate economy, is encouraging. Birth-dose vaccination has been advocated in Bangladesh, but has not been incorporated into a national program. Instead, Bangladesh is trying to maintain the ongoing EPI, because there is no infrastructure for birth-dose vaccination. The integration of HepB into a pentavalent vaccine has received approval from the people. There are many important players in vaccine promotion in developing countries, and HepB as a ‘partner’ of pentavalent vaccine seems to be a feasible approach. The other three SEARO countries with successful HB vaccination programs report similar findings. By contrast, while India introduced birth-dose vaccine to contain HBV, the vaccine coverage is unsatisfactory, and significant heterogeneity in vaccine coverage among states and tribes in India remains. However, the elimination of hepatitis by 2030 may not be achieved with existing approaches. The original target was to achieve less than 1% HBsAg-positivity in children aged less than 5 years by 2020. Now, attention should focus on reducing HBsAg positivity to below 0.1% within the next 10 years (by 2030). To achieve this, the capacity and quality of the health care delivery system of individual countries should be properly analyzed. The cost and infrastructure facility to implement of a birth dose HBV vaccination versus increasing the coverage of HepB3 should be critically analyzed at local levels. Subsequently, antenatal care, estimation of HBV DNA and HBeAg, and the use of antivirals in mothers with increased viral loads and HBeAg-positivity should all be considered. Implementation of HepB-BD should afterwards be considered critically by analyzing the prevailing vaccination programs. In conclusion, proper execution of HepB-BD, HepB3, doses, antenatal care and development of insights about prevalence of HBV variants would allow the achievement of the target of HB vaccination in 2020 and 2030, however, different countries might prioritize different vaccination schedules, and there might even be considerable heterogeneity in policy implementation in different places of some countries.

## Figures and Tables

**Table 1 vaccines-09-00374-t001:** Hepatitis B vaccine schedule and estimated coverage [19].

Country	HepB3 Year of Introduction	HepB Schedule	HepB3 Coverage (2019)	Birth Dose Year of Introduction	HepB-BD Coverage (2019)
Bangladesh	2003	6, 10, 14 weeks	98%	ND	NA
Bhutan	1997	0, 6, 10, 14 weeks	97%	2012	86%
India	2002	0, 6, 10, 14 weeks	91%	2011	56%
Indonesia	1997	0, 2, 3, 4, 18 months	85%	2002	84%
Maldives	1993	0, 8, 12, 24 months	99%	2000	99%
Myanmar	2003	0, 8, 12, 24 months	90%	2016	17%
Nepal	2002	6, 10, 14 weeks	93%	ND	NA
North Korea	2003	0, 6, 10, 14 weeks	97%	2004	98%
Si Lanka	2003	8, 16, 24 months	99%	ND	NA
Thailand	1992	0, 2, 4, 6 months	97%	1992	99%
Timor-Leste	2007	0, 6, 10, 14 weeks	83%	2007	70%

HepB3, three doses of hepatitis B vaccine; HepB-BD, birth dose of monovalent hepatitis B vaccine; ND, not done; NA, not available.

**Table 2 vaccines-09-00374-t002:** Kinetics of HBsAg seroprevalence in four countries that met the WHO target of containing HBV in children *.

Country	HBsAg Seroprevalence before Vaccine Introduction % (95% CI)	HBsAg Seroprevalence in Children Aged ≥5 Years after Vaccine Introduction % (95% CI)	Year of Verification of < 1.0% HBsAg Seroprevalence
Bangladesh	1.2 (0.7–1.6%)	0.05 (0.0–0.01%)	2019
Bhutan	2.0 (1.0–4.0%)	0.5 (0.1–1.8%)	2019
Nepal	0.3 (0.1–09%)	0.01 (0.04–0.4%)	2019
Thailand	4.5%	0.3%	2019

* Born after implementation of the national universal hepatitis B infant immunization program.

## Data Availability

Data will be available from references. The corresponding author will also supply if any data is required.

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
