# Peer review of "Implications of Birth-Dose Vaccination against Hepatitis B Virus in Southeast Asia"

_vaccines, 2021, doi:10.3390/vaccines9040374_

Round 1

Reviewer 1 Report

The review entitled "Implications of birth-dose vaccination against hepatitis B virus  in Southeast Asia" covers hepatitis B vaccination target by SEARO countries, implications of HBV vaccine for SEARO countries at birth date, the role of vaccination in prevention HBV tranmission to newborns,  and resons of missing some SEARO countries the hepatitis B vaccination target.

The review is short in its content and it needs to be expanded to include important items related to HBV vaccinations such as

a) HBV vaccine esacape mutation in SEARO: in this point the authors should provide the following information

  i) What at the vaccine escape mutants in SEARO?

  ii) Correlate the mutants with HBV vacciantiona in SEARO?

  iii) Correlate the mutants with HBs Ag  as in table 2

b) Does the vaccine reduce the risk rate of mutations appeared in HBV polymerase during therapy?

I would suggest the authors to make another table including countries, vaccine escape mutants appeared, prevalence of mutation with time (either increase or decrease) .

Author Response

Comments and Suggestions for Authors

The review entitled "Implications of birth-dose vaccination against hepatitis B virus in Southeast Asia" covers hepatitis B vaccination target by SEARO countries, implications of HBV vaccine for SEARO countries at birth date, the role of vaccination in prevention HBV transmission to newborns, and reasons of missing some SEARO countries the hepatitis B vaccination target.

The review is short in its content and it needs to be expanded to include important items related to HBV vaccinations such as:

Response: Thanks for your understandings. We have added a new chapter to respond to your queries. Other points have been discussed as per your suggestions.

Query

  1. a) HBV vaccine escape mutation in SEARO: in this point the authors should provide the following information
  2. i) What at the vaccine escape mutants in SEARO?
  3. ii) Correlate the mutants with HBV vaccination in SEARO?

  iii) Correlate the mutants with HBs Ag as in table 2

Response

A new chapter entitled “HBV vaccine escape mutation” has been formulated and introduced in revised manuscript (Chapter 7). In Table 2, data of 4 countries have been provided. Five new references have been incorporated in the revised manuscript to validate about HBV vaccine escape mutation (Reference 24-28). Regarding Table 2, there has been no data about HBV vaccine escape of Bhutan and Nepal. Little is known about HBV vaccine escape mutation in Bangladesh and Thailand. The finding has been discussed, but it cannot be tabulated due to lack of convincible data..

Query:

  1. b) Does the vaccine reduce the risk rate of mutations appeared in HBV polymerase during therapy?

Response

There are some evidences that show that vaccine may reduce the risk of polymerase therapy. In addition, vaccine may also be used for treatment of patients with chronic hepatitis B.

In chimpanzee, it has been shown that HB vaccination provide protection against evolving of mutant during polymerase therapy. (Kamili S, Sozzi V, Thomson G, Campbell K, Walker CM, Locarnini S, Krawcyznski. Efficacy of hepatitis B vaccine against antiviral drug-resistant hepatitis B virus mutants in the chimpanzee model. Hepatology 2009 May;49(5):1483-91. doi: 10.1002/hep.22796. PMID: 19274751. DOI: 10.1002/hep.22796)

In human therapeutic vaccines containing HBsAg and other HBV-related proteins reduce viral replication and induce normalization of alanine aminotransferase (ALT) when these are given in patients with chronic hepatitis B. In addition, we and other have used HBsAg/HBcAg-based therapeutic vaccine with nucleoside analogs to treat patients with CHB. This treatment has achieved functional cure in some patients (29-31; Al-Mahtab M, Akbar SM, Aguilar JC, Uddin MH, Khan MS, Rahman S. Therapeutic potential of a combined hepatitis B virus surface and core antigen vaccine in patients with chronic hepatitis B. Hepatol Int. 2013;7(4):981-9.

Al Mahtab M, Akbar SM, Aguilar JC, Guillen G, Penton E, Tuero A, Yoshida O, Hiasa Y, Onji M. Treatment of chronic hepatitis B naïve patients with a therapeutic vaccine containing HBs and HBc antigens (a randomized, open and treatment controlled phase III clinical trial). PLoS One.13(8):e0201236. (2018) doi: 10.1371/journal.pone.0201236. eCollection 2018. PMID:30133478

Horiike N, Akbar SM, Michitaka K, Joukou K, Yamamoto K, Kojima N, Hiasa Y, Abe, Onji M. In vivo immunization by vaccine therapy following virus suppression by lamivudine: a novel approach for treating patients with chronic hepatitis B. J Clin Virol 2005; 32: 156-161). However, the role of prophylactic vaccine in regulating development of mutant HBV during nucleoside treatment is not clear.)

Query: I would suggest the authors to make another table including countries, vaccine escape mutants appeared, prevalence of mutation with time (either increase or decrease)

Response: Reliable data are not available to construct a Table of this nature.

Reviewer 2 Report

This is a useful review of a key issue in regard to HepB vaccination. It draws together information on the context of the SEARO countries, while focusing on key issues in terms of HepB vaccination. 

The paper is clearly structured, and makes good use of available data and published literature. 

My main suggestion is to 'sharpen' the focus of the paper by avoiding reference to 'pros and cons' - which is rather vague - and focusing more on the key question of future directions for the HepB vaccination program in SEARO countries. 

Specifically:

(1) Abstract provides the context but little on the content. For example, rather than ‘examines the pros and cons’ – reviews the experience of SEARO countries in control of hepB and describes options for strengthening / expanding the hepB vaccination program

(2) Summary: Suggest reword as 'conclusions' -could this section be strengthened to indicate the key issues for consideration (eg HepB birth dose, HepB 3 doses, ANC HepB testing and treatment) and the contextual issues that need to be considered for each option ?  

More detailed comments follow: 

Line 63 ‘birth dose combats HBV more efficiently’ but no reference for this. Unclear whether efficiently means at less cost, or more effectively. Provide more clarification and reference for this statement. Thus, rather than a review of 'pros and cons', the review identifies the key options, and provides some guidance as to the context in which each option could be considered.

Line 169 recommendations on ANC by HBsAg testing and antiviral treatment – the effectiveness and costs of these recommendations compared to improving HB3 coverage needs to be considered. Given limited resources what would provide the most reduction in HBV prevalence for the least cost ?

Line 194 ff see comment above. Refers again to ‘pros and cons’ of ongoing vaccination programs. This is not a very clear recommendation. Suggest provide a more precise recommendation – that incorporates a review of the health system capacity and coverage, the epidemiology of HBV, particularly prevalence in pregnant women, and the relative feasibility and costs of including a birth dose of HepB vaccine, vs increasing the coverage of HepB3. Given variations in epidemiology and health system capacity among SEARO countries, different countries might prioritize different vaccination schedules – and even within countries (especially large countries such as India or Indonesia), regional variation in schedules could be appropriate.

Author Response

Response to Reviewer 2

Thanks for your kind review and constructive suggestions.

This is a useful review of a key issue in regard to HepB vaccination. It draws together information on the context of the SEARO countries, while focusing on key issues in terms of HepB vaccination. 

The paper is clearly structured, and makes good use of available data and published literature. 

My main suggestion is to 'sharpen' the focus of the paper by avoiding reference to 'pros and cons' - which is rather vague - and focusing more on the key question of future directions for the HepB vaccination program in SEARO countries. 

Specifically:

Query

  • Abstract provides the context but little on the content. For example, rather than ‘examines the pros and cons’ – reviews the experience of SEARO countries in control of hepB and describes options for strengthening / expanding the hepB vaccination program

Response

The wording of the sentence has been changed as per recommendation of the Reviewer Abstract. Last 3 lines.

  • Summary: Suggest reword as 'conclusions' -could this section be strengthened to indicate the key issues for consideration (eg HepB birth dose, HepB 3 doses, ANC HepB testing and treatment) and the contextual issues that need to be considered for each option?  
  • Response: The “Summary has been revised as recommended” (Line 293 to 298; Line 300 to line 304)

More detailed comments follow: 

Query: Line 63 ‘birth dose combats HBV more efficiently’ but no reference for this. Unclear whether efficiently means at less cost, or more effectively. Provide more clarification and reference for this statement. Thus, rather than a review of 'pros and cons', the review identifies the key options, and provides some guidance as to the context in which each option could be considered.

Response: Reference to support the statement has been given as Reference 10, 11. Also, the benefits of Birth-dose HBV vaccination have been explained. Additionally, the concept of the study has been shown in revised manuscript. The efficiency and cost-effectiveness of HepB-BD has been shown in the text (Line 70-86).

Query: Line 169 recommendations on ANC by HBsAg testing and antiviral treatment – the effectiveness and costs of these recommendations compared to improving HB3 coverage needs to be considered. Given limited resources what would provide the most reduction in HBV prevalence for the least cost?

Response: The query has been responded (Line 257-269).

Query:  ff see comment above. Refers again to ‘pros and cons’ of ongoing vaccination programs. This is not a very clear recommendation. Suggest provide a more precise recommendation – that incorporates a review of the health system capacity and coverage, the epidemiology of HBV, particularly prevalence in pregnant women, and the relative feasibility and costs of including a birth dose of HepB vaccine, vs increasing the coverage of HepB3. Given variations in epidemiology and health system capacity among SEARO countries, different countries might prioritize different vaccination schedules – and even within countries (especially large countries such as India or Indonesia), regional variation in schedules could be appropriate.

Response: The query has been addressed as per recommendation (Line 292-304).

Reviewer 3 Report

HBV active infection is one of the most important public health in the word. Although the routes of the transmission are different, however vertical transmission is of particular importance in endangering the health of the newborn. I find the publication interesting and informative about vaccination program in selected countries in Asia. However I suggest continuing the research on HBV infection among pregnant women and how the situation changed after vaccination program implementation. I am sure that such comprehensive analysis and statistical models will give more information what is the real epidemiological threat of HBV. The presented paper should be published.

Author Response

Response to Reviewer 3

HBV active infection is one of the most important public health in the word. Although the routes of the transmission are different, however vertical transmission is of particular importance in endangering the health of the newborn. I find the publication interesting and informative about vaccination program in selected countries in Asia. However I suggest continuing the research on HBV infection among pregnant women and how the situation changed after vaccination program implementation. I am sure that such comprehensive analysis and statistical models will give more information what is the real epidemiological threat of HBV. The presented paper should be published.

Response: Thank you very much for your inspiring comments and we would continue to work on this field.

Round 2

Reviewer 1 Report

The review is improved significantly after addition of vaccine escape mutants. The review could be published now.

Author Response

Response to Reviewer 1

Comments of the Reviewer:

The review is improved significantly after addition of vaccine escape mutants. The review can be published no.

Response: Thanks for your understandings;

Reviewer 2 Report

Responses To Reviewer comments

(1) Abstract provides the context but little on the content. For example, rather than ‘examines the pros and cons’ – reviews the experience of SEARO countries in control of hepB and describes options for strengthening / expanding the hepB vaccination program

Response

The wording of the sentence has been changed as per recommendation of the Reviewer Abstract. Last 3 lines.

Response is satisfactory with the following minor edit: (changes in bold)

The article discusses options for strengthening and expanding the Hepatitis B vaccination program in SEARO countries with an emphasis on HepB and HepB-BD programs.

(2) Summary: Suggest reword as 'conclusions' -could this section be strengthened to indicate the key issues for consideration (eg HepB birth dose, HepB 3 doses, ANC HepB testing and treatment) and the contextual issues that need to be considered for each option?

Response: The “Summary has been revised as recommended” (Line 293 to 298; Line 300 to line 304)

Response is satisfactory with the following Minor editing: (changes in bold and deletions in track changes)

To achieve this, the capacity and quality of the health care delivery systems of individual country should be properly analyzed. The cost and infrastructure facility capacity to implement of a birth dose HBV vaccination versus increasing the coverage of HepB3 should be critically analyzed at local levels. Subsequently, antenatal care, estimation of HBV DNA and HBeAg, and the use of antivirals in mothers with increased viral loads and HBeAg-positivity should all be considered. Implementation of  HepB-BD should be considered after critically analyzing the prevailing vaccination programs. In conclusion, proper execution of HepB-BD and HepB3 doses, antenatal care for HBV positive mothers, and development of insights about prevalence of HBV variants would allow achievement of the target of HB vaccination in 2020 and 2030. However, different countries might prioritize different vaccination schedules and there might considerable heterogeneity in policy implementation in different places of some countries.

More detailed comments follow:

(3) Query: Line 63 ‘birth dose combats HBV more efficiently’ but no reference for this. Unclear whether efficiently means at less cost, or more effectively. Provide more clarification and reference for this statement. Thus, rather than a review of 'pros and cons', the review identifies the key options, and provides some guidance as to the context in which each option could be considered.

Response: Reference to support the statement has been given as Reference 10, 11. Also, the benefits of Birth-dose HBV vaccination have been explained. Additionally, the concept of the study has been shown in revised manuscript. The efficiency and cost-effectiveness of HepB-BD has been shown in the text (Line 70-86).

Response is satisfactory with the following  minor edits.

Minor edits

line 85: replace backgrouns with backgrounds

Line 97 add ‘the’ before ‘overall target’ and omit ‘in these countries’ repeated at end of sentence.

line 102: replace obstacle with obstacles

Line 103: replace ‘about traditional Hep B and HepB BD in SEARO countries’ with ‘on current HepB and HepB BD programs in SEARO countries’

Line 189; replace SREAO with SEARO;

Line 191: omit ‘and realized’; replace ‘birth diose’ with ‘birth dose’

Line 191-194: replace ‘Also, there should be clear explanation regarding future doses of hepatitis B vaccination those should be taken from different institutions. If these can not be ascertained, there will be a negative impact on HBV vaccination program conducted by EPI.’

With ‘Mothers will also require clear information on the need for future doses of HepB vaccine for their infant, and on how to access facilities to obtain these doses. Without this information, there is a risk of a negative impact on overall HBV vaccination coverage through the EPI program’.

(4) Query: Line 169 recommendations on ANC by HBsAg testing and antiviral treatment – the effectiveness and costs of these recommendations compared to improving HB3 coverage needs to be considered. Given limited resources what would provide the most reduction in HBV prevalence for the least cost?

Response: The query has been responded (Line 257-269).

The response addresses the query well.

(5) Additional paragraphs 6 on data limitation and 7 on HBV escape mutation are useful additions. However, checking and correction of grammar, syntax and spelling during minor editing is required.

(7) Query: Refers again to ‘pros and cons’ of ongoing vaccination programs. This is not a very clear recommendation. Suggest provide a more precise recommendation – that incorporates a review of the health system capacity and coverage, the epidemiology of HBV, particularly prevalence in pregnant women, and the relative feasibility and costs of including a birth dose of HepB vaccine, vs increasing the coverage of HepB3. Given variations in epidemiology and health system capacity among SEARO countries, different countries might prioritize different vaccination schedules – and even within countries (especially large countries such as India or Indonesia), regional variation in schedules could be appropriate.

Response: The query has been addressed as per recommendation (Line 292-304).

The response is satisfactory and provides some clearer recommendations for further action. However there are a number of grammatical errors which require minor editing:

Line 292  add ‘the’ before health care delivery system

Line 293 convert country to ‘countries’

Line 294 to implement of – delete of; correct does to ‘dose’

Line 298 omit ‘by’ before analyzing

Line 299 omit ‘doses’

Line 300 convert ‘to achieve’ to ‘achievement of’

Line 301 covert comma to full stop after 2030. Then capital However,

Line 302 omit ‘even’, change to ‘there might be considerable..’

Author Response

Response to Revewer-2

Comment of the Reviewer:

The response is satisfactory and provides some clearer recommendations for further action. However, there are a number of grammatical errors which require minor editing:

Line 292 add ‘the’ before health care delivery system

Response: This has been corrected

Line 293 convert country to ‘countries’

Response: This has been corrected

Line 294 to implement of – delete of; correct does to ‘dose’

Response: This has been corrected

Line 298 omit ‘by’ before analyzing

Response: This has been corrected

Line 299 omit ‘doses’

Response: This has been corrected

Line 300 convert ‘to achieve’ to ‘achievement of’

Response: This has been corrected

Line 301 covert comma to full stop after 2030. Then capital However,

Response: This has been corrected

Line 302 omit ‘even’, change to ‘there might be considerable..’

Response: This has been corrected
